# Heat Shock Protein 22 in Physiological and Pathological Hearts: Small Molecule, Large Potentials

**DOI:** 10.3390/cells11010114

**Published:** 2021-12-30

**Authors:** Xiaonan Sun, Sharadhi Siri, Amirah Hurst, Hongyu Qiu

**Affiliations:** Center for Molecular and Translational Medicine, Institute of Biomedical Science, Georgia State University, Atlanta, GA 30303, USA; xsun13@gsu.edu (X.S.); ssiri1@student.gsu.edu (S.S.); ahurst3@gsu.edu (A.H.)

**Keywords:** HSP22, cardiomyopathy, heart, cardiac hypertrophy, myocardial ischemia, aging

## Abstract

Small heat shock protein 22 (HSP22) belongs to the superfamily of heat shock proteins and is predominantly expressed in the heart, brain, skeletal muscle, and different types of cancers. It has been found that HSP22 is involved in variant cellular functions in cardiomyocytes and plays a vital role in cardiac protection against cardiomyocyte injury under diverse stress. This review summarizes the multiple functions of HSP22 in the heart and the underlying molecular mechanisms through modulating gene transcription, post-translational modification, subcellular translocation of its interacting proteins, and protein degradation, facilitating mitochondrial function, cardiac metabolism, autophagy, and ROS production and antiapoptotic effect. We also discuss the association of HSP22 in cardiac pathologies, including human dilated cardiomyopathy, pressure overload-induced heart failure, ischemic heart diseases, and aging-related cardiac metabolism disorder. The collected information would provide insights into the understanding of the HSP22 in heart diseases and lead to discovering the therapeutic targets.

## 1. Introduction

The heat shock proteins (HSPs) are a family of proteins that have been linked to different cellular functions in multiple tissues. Although HSPs were named due to the discovery of their response to high temperatures in the cells, they have since been observed to be activated under diverse cellular stresses, such as toxins, radiation, infectious agents, hypoxia, oxidative stress, etc. Numerous studies have shown that HSPs play a critical role in maintaining protein and cellular homeostasis and protect cells against injury by regulating protein folding, intracellular protein trafficking, and responding to unfolded and denatured proteins resulting from the stressors [1,2,3]. Thus, most of the HSPs were classified as stress-induced proteins due to their rapid and high inducibility in response to stress, while fewer HSPs were considered as stress-independent proteins due to their constitutive expression in cells.

HSPs possess molecular sizes ranging from 10 to 150 kDa and are characterized according to their molecular weight [4]. The small HSPs are a group of HSPs, 12–43 kDa in size, containing a core α-crystallin domain (ACD) with variable N-terminal and C-terminal domains. In humans, ten ACDs have been identified for the small HSPs, including HSPB1, HSPB2, HSPB3, HSPB4, HSPB5, HSPB6, HSPB7, HSPB8, HSPB9, and HSPB10 [5]. In general, small HSPs function as a chaperone to stabilize injured proteins and prevent misfolded protein interactions and aberrant protein aggregation by exposing the hydrophobic residues of these proteins [3,6,7].

HSP22, also known as H11 kinase or HSPB8, is one of the small HSP family members [6,8]. HSP22 was initially found to be highly expressed in various tumors, such as gastric cancer [9], breast cancer [10], ovarian carcinoma [11], and hepatocellular carcinoma [12] and to be involved in the regulation of cell growth and apoptosis, thus participating in carcinogenesis. Point mutants of HSP22 were also found to be correlated with the development of different neuromuscular diseases [13,14]. A recent study has also shown a protective effect of HSP22 in ischemic lung injury [15]. In addition, previous studies showed that HSP22 is predominately expressed in a limited number of tissues under physiological conditions, including heart and skeletal muscle, and is involved in the regulation of cell growth, leading to cardiac hypertrophy [6,16]. Subsequent evidence has indicated that HSP22 was also induced in the heart tissues under various conditions of myocardial ischemic stress in animal models and patients [17,18,19]. More importantly, it has been shown that overexpression of HSP22 protects cardiomyocytes against oxidative stress-induced cell death in vitro and reduces myocardial ischemic injury in vivo [20,21,22,23]. Another recent study also showed that HSP22 suppresses diabetes-induced endothelial injury [24]. Reciprocally, the knockdown of HSP22 increases stress-induced cardiomyocyte death and accelerates the transition into heart failure [25]. One more recent study also showed that the loss of HSP22 progressively induced cardiac dilation and dysfunction [26]. All these studies have highlighted the importance of HSP22 in the heart under both physiological and pathological conditions. The multifunctionality of HSP22 has been highlighted as a reason why it is a reliable target for therapeutic purposes in various heart diseases.

In this review, we focus on the studies of multiple functions of HSP22 in the heart, outlining the associated molecular mechanisms involved in cytoprotection of cardiomyocytes, and its modulation in autophagy, mitochondrial function, energy metabolism, and oxidative stress. The molecular mechanisms include the regulations in protein expression and activation of pre-survival signaling, subcellular protein translocation, and protein degradation via activating proteasome. We also discuss the discovery of the protective role of HSP22 against various cardiac stress, such as diabetes, ischemia, pressure overload, and aging. The summarized information would further increase our knowledge of the principle of HSP22 in the heart and open up novel therapeutic possibilities for heart diseases. It would also bring a fresh perspective into understanding the role of HSP22 in other conditions, such as cancers, neurological disorders, and age-related diseases.

## 2. The Molecular Mechanisms of HSP22’s Cytoprotection in Cardiomyocytes

HSP22 is comprised of 196 amino acids and has an α-crystallin domain next to the C-terminus [6,8]. It has been demonstrated that HSP22 possesses a highly flexible structure and tends to form small-molecular-mass oligomers. As a stress-associated protein, HSP22 was found to be induced in the heart tissues under various cell-extrinsic and intrinsic stresses. Although the upstream activators of HSP22 under different stresses are far from being fully understood, studies have shown that the expression of HSP22 under stressed conditions is largely regulated by the upregulated transcription of heat shock factors (HSF). Under normal conditions, the expression of HSF is highly conserved when interacting with HSPs and the activity of HSF is maintained in an inactive state. Under stress conditions, the unfolded protein may disrupt the interaction between HSP22 and HSF, which induces the activation and translocation of HSF. By binding with heat shock elements (HSEs) in the HSP22 gene promoter, the transcription of HSP22 occurs. HSFs are also regulated by several post-translational modifications like phosphorylation, acetylation and sumoylation [27,28]. The expression and activation of HSF1 and HSF2 are involved in heart development and regulate the expression of HSPs in cardiovascular diseases [29,30]. In particular, the HSF1 binding site is upstream of the HSP22 translation start site [16]. Studies have also shown that the expression and activity of HSP22 are regulated by its interacting proteins in various cells. For example, in the heart, the function of HSP22 is highly dependent on a co-chaperone protein: BCL-2–associated athanogene 3 (BAG3). With the deletion of BAG3, the expression of HSP22 is also eliminated [31]. In addition, HSP22 can be phosphorylated by protein kinase C and plays a role in the maintenance of smooth muscle integrity by interacting with HSP27 [8].

Although the physiological function of HSP22 is still largely unknown, evidence indicates that HSP22 might play distinct roles in a dose-dependent manner [32]. For instance, HSP22 has been found to increase cell size at low doses while promoting cell death at higher doses in cardiomyocytes [19]. Meanwhile, HSP22 triggered both cell growth and cell survival by affecting different signaling pathways, e.g., HSP22 promotes cardiac hypertrophy via activating phosphoinositide 3-kinases (PI3K)–Akt signaling but induces apoptosis by inhibiting casein kinase 2 (CK2) activity [19]. Such varied functions of HSP22 give scientists insights into intracellular protective mechanisms in cardiomyocytes and the defense mechanism against cell injury in the heart. Here, we summarize the fundamental molecular mechanisms of HSP22 involved in the protection of cardiomyocytes.

### 2.1. HSP22 Activates Multiple Cardiac Pro-Survival Signaling

Previous studies from human and animal models showed that HSP22 was significantly induced in the myocardium after acute and chronic ischemia. Specifically, HSP22 was increased in the stunning and hibernating myocardium [17,18,19], indicating an adaptive response to oxygen deprivation. Thus, HSP22 was assumed to be a cytoprotective protein preventing irreversible ischemic damage of the stunning and hibernating cardiomyocytes. Further studies during the past years have revealed the protective effects of HSP22 in the heart and discovered novel molecular mechanisms by which HSP22 regulates the pro-survival signaling pathways in the cardiomyocytes. As summarized in a recent review article [33], in vitro studies with isolated myocytes revealed that overexpression of HSP22 protected cardiomyocyte against stress-induced cell apoptosis by activating PI3K/Akt/Smad 1/5/8 pro-survival signaling [22]. The results showed that overexpression of HSP22 in isolated neonatal cardiomyocytes significantly increased PI3K activity and Akt and Smad 1/5/8 phosphorylation. Additionally, a previous study showed that cardiac-specific overexpression of HSP22 significantly increased the phosphorylation of signal transducer and activator of transcription 3 (STAT3) on S727 when compared with wild type (WT) mice, while silencing Hsp22 in isolated neonatal rat cardiomyocytes attenuated STAT3 activation [25]. Furthermore, several studies showed a tight association between the HSP22 and inducible nitric oxide synthase (iNOS) and revealed that HSP22-mediated antiapoptotic effect in cardiomyocytes relies on its induction of iNOS [20,21]. Overexpression of HSP22 increased iNOS expression and activity in cardiomyocytes and heart tissues, and deletion of, or silencing HSP22 attenuated iNOS expression. Importantly, inhibition of iNOS abolished HSP22-induced cardioprotection [20,21]. It was found that HSP22 increased an AAA-associated protein named valosin-containing protein (VCP) in the Hsp22 transgenic (TG) mouse. HSP22 co-located with AKT and VCP [34] and interacted with these two proteins, predominated in the nuclear fraction of cardiac myocytes, which in turn induced iNOS expression in the cardiomyocytes through the activation of the transcription factor nuclear factor kappa-light-chain-enhancer of activated B cells (NF-kB) [34]. It is notable that although a moderate increase in iNOS (2- to 3-fold) has been demonstrated to be beneficial, excessive iNOS may be toxic to the cardiomyocytes. It is also important to control the iNOS increase at a moderate level. Moreover, HSP22 also increased the association of aurora-like kinase (Alk3) and the bone morphogenetic protein receptor type II (BMPR-II), and it interacted with the transforming growth factor-beta-activated kinase (TAK)1, mediating the BMP receptor signaling [22]. These results suggest that HSP22 mediates multiple survival signaling in cardiomyocytes.

### 2.2. HSP22 Chaperones Its Client Proteins and Facilitates Their Subcellular Redistribution

Studies showed that HSP22 also participated in the subcellular redistribution of other proteins, contributing to its diverse cellular function. HSP22 predominately accumulates in the perinuclear compartment and inner membrane of the mitochondria, which provides a fundamental basis for chaperoning or interacting with the other proteins into nucleus or mitochondria [35,36]. For example, HSP22 promotes the peri-nuclear accumulation of Akt and VCP, resulting in the activation of the transcription factor NF-kB, which in turn induces the expression of iNOS [25,34]. On the other hand, HSP22 was also located in the mitochondria of cardiomyocytes and played a regulatory role in mitochondrial respiration in an iNOS-dependent manner [35]. HSP22 promotes the translocation of STAT3 and iNOS to the mitochondria, regulating mitochondrial function in cardiomyocytes [25,35]. It has also been shown that the mitochondrial translocation of HSP22 relies on its N-terminal domain, resulting in translocation of iNOS to mitochondria, thus playing a role in mitochondrial function and cytoprotection [35]. Deleting the N-terminal domain of HSP22 blocks its mitochondrial translocation, attenuates iNOS mitochondrial distribution, and subsequently impairs oxidative phosphorylation, preventing cytoprotection [35]. These findings are important since cardioprotection by NO donors has been limited so far; the observed protective effects of iNOS induced by HSP22 imply that stimulating endogenous iNOS mitochondrial translocation inside the cardiac myocytes might have better biological efficiency than that provided by NO donors.

### 2.3. HSP22 Enhances Proteasome Activity and Participates in Energy Metabolism

It has also been shown that HSP22 involved in the proteasome-related cell growth in cardiomyocytes. The ubiquitin–proteasome system (UPS), degrading about 80% of the intracellular proteins, represents the most important mechanism of proteolysis in the cardiac cell. A study showed that overexpression of HSP22 in a TG mouse reveals an increased expression of the proteasome in hearts in both 19S and 20S subunits and the 20S catalytic activity. Inhibition of proteasome activity by epoxomicin reduced hypertrophy in TG mice. These results were confirmed in HSP22 over-expressed isolated cardiac myocytes [37]. The data from these studies indicated that HSP22-mediated cardiac hypertrophy promotes increased expression, activity, and subcellular redistribution of the proteasome. In addition, HSP22 was also found to be involved in the metabolic switch in the ischemic heart by promoting the activity of 5′ AMP-activated protein kinase (AMPK), which subsequently stimulates glucose uptake and glycolysis to compensate for the lack of aerobic ATP production in the ischemic heart [20,36]. A most recent study showed that loss of HSP22 impaired fatty acid (FA) and glucose metabolism by interfering with the critical regulatory enzymes in FA transport and FA oxidation (FAO) and with glycolysis and gluconeogenesis, respectively, undermining ATP production in physiological aging transition, leading to cardiac dilation and dysfunction [26].

### 2.4. HSP22 Participates in Cardiac Autophagy via BAG3

HSP22 has also been reported to regulate cardiac autophagy. HSP22, along with the co-chaperone BAG3, plays a key role in the autophagic degradation in the heart, and this process is termed chaperone-assisted selective autophagy [38]. Previous studies have revealed a functional dependency of HSP22 on BAG3 [39,40,41], which is a co-chaperone protein abundantly expressed in the heart. BAG3 deletion eliminates HSP22 expression in cardiomyocytes [31]. BAG3 mutations or deletion have been associated with the development of human dilated cardiomyopathy (DCM) [42,43]. BAG3 has been reported to act in concert with HSP22 to induce autophagic degradation, especially in cardiomyocytes [20]. On the other hand, a study showed that deletion of HSP22 reduced the expression of BAG3 and impaired cardiac autophagy in the aging heart [26].

### 2.5. HSP22 Regulates Cardiac Reactive Oxygen Species (ROS) Production and Oxidative Stress

Furthermore, research has also indicated that HSP22 regulates ROS production and oxidative stress in the cardiomyocytes. In vitro studies have shown that overexpression of HSP22 in isolated cardiomyocytes significantly reduces H_2_O_2_-mediated apoptosis. In an ischemic heart, with direct binding to Akt and AMPK, HSP22 induces the survival mechanisms by the inhibition of pro-apoptotic factors and the activation of anti-apoptotic factors [20,21,22,23]. In addition, HSP22 overexpression attenuates ROS production in isolated mitochondria from the cardiomyocytes from HSP22 TG mice. Loss of HSP22 results in age-dependent accumulation of oxidative cytotoxic products in the heart, leading to cardiac oxidative damage in older HSP22 KO mice [26]. Paradoxically, studies also showed that, under the normoxia condition, overexpression of HSP22 increases ROS production, which may be associated with the increased activity and expression of NADPH oxidase 2 (Nox2) and the enhanced activity of xanthine oxidase in the myocardium [44]. These data indicate that the role of Hsp22 under the physiological condition differs from that under the oxidative stress in the heart and further supports the concept that Hsp22 is a stress-associated protein in the heart that has a distinct protective effect under oxidative stress. These opposite observations [26,44] may also be due to the difference in the expression level and affected period of HSP22. For example, moderate increase and short-term effect of HSP22 result in a reduction in ROS and attenuation of oxidative stress, such as in isolated culture cells [22] or young mouse heart [26], while the extremely high-level expression and long-term chronic effect of HSP22 increases ROS production, as observed in aged TG mice [44]. These data indicate that the role of HSP22 in ROS varies depending on the heart condition and needs more investigation. In summary, these results suggested a comprehensive function of HSP22 in cardiomyocytes offering cardiac protection. Hsp22 mediates a concomitant activation of several survival kinases and creates a cross-talk network among these molecules [22,33,34]. As a chaperone protein, Hsp22 interacts with multiple molecules, such as AKT, STAT3, INOS, and VCP, and facilitates the subcellular redistribution of these interacting proteins, regulating gene expression, post-translational modification, and mitochondrial functions [22,33,34]. In addition, Hsp22 was involved in the metabolic switch during stress and energy production. Furthermore, Hsp22 enhances the expression and activity of the proteasome and participates in cardiac autophagy and ROS production. All these studies have highlighted the importance of HSP22 in regulating the cellular function of cardiomyocytes. The associated signaling pathway and cellular processes are illustrated in Figure 1.

## 3. The Association of HSP22 in Heart Diseases

### 3.1. HSP22 in Human DCM

DCM is one of the leading causes of heart failure (HF) characterized by left ventricular (LV) enlargement and contractile dysfunction [42]. Clinically, DCM occurs in the absence of obvious etiology, such as coronary artery disease, hypertension, and congenital heart disease [45]. In 20–48% of the cases, the condition is genetically inherited, principally caused by mutations in genes that encode for sarcomeric and cytoskeletal proteins in the cardiac myocyte or due to imbalances in calcium homeostasis [46]. It affects approximately 1 in 2500 people and is more common in men above 55 than in women and younger populations [47]. DCM is histologically characterized by myocyte hypertrophy along with diffuse fibrosis.

Although the function of HSP22 in cardiac cell growth and cardioprotection is increasingly being studied, there is little work on the role HSP22 plays in human dilated cardiomyopathy. One study showed that single point mutations in the HSPB8 gene, which encodes for HSP22, such as K141N, can cause cardiomyopathy. HSPB8 K141N TG mice showed mild hypertrophy, fibrosis, and reduced cardiac function [48]. Interestingly, the dose-dependent effect of HSP22 was observed, as increased levels of the protein in the double TG mice contributed to cellular toxicity and exhibited a stronger cardiomyopathy phenotype [36]. Studies have shown that recombinant HSP22 can interrupt amyloid formation, thus proving to be a potential therapeutic strategy for treating specific cardiomyopathies [49]. In addition, mutated or deficient BAG3 expression has been repeatedly associated with the development of human cardiomyopathy [42,43], which is usually accompanied by the reduction in HSP22 levels. Loss of HSP22 also reduced BAG3 expression and resulted in alterations in the heart similar to human DCM [26]. These results together indicate a potential link of HSP22 with the development of human DCM. However, more clinical evidence is needed from future investigations.

Based on these results, researchers found HSP22 shows a dose-dependent dual function, as lower doses resulted in increased cell size and higher doses showed proapoptotic activity. It was supposed that at lower doses, HSP22 interacts with Akt in a kinase-independent manner, while at higher doses, HSP22’s apoptotic activity occurs through a protein kinase-dependent pathway. The latter was said to occur through the binding and inhibition of casein kinase 2, which in itself exhibits proapoptotic activity [14]. Additionally, it was observed that the K113G mutant exhibited no proapoptotic activity but induced cardiac hypertrophy. This may be because the K113G mutant of HSP22 does not bind casein kinase 2 as opposed to the WT Hsp22 [19]. There have been contradictory conclusions drawn from these results, as certain groups disagree with the “dose-dependent dual function” attributed to Hsp22. Thus, the intrinsic protein kinase activity of Hsp22 is unclear, and it may result from the interaction with other proteins in the cell [32].

### 3.2. HSP 22 in Pressure Overload or Hypertensive Cardiac Hypertrophy and HF

Hypertensive cardiac hypertrophy is characterized by cardiac remodeling and increased cardiomyocyte death, which leads to reduced systolic and diastolic function, ultimately resulting in HF. Cardiac hypertrophy can develop with either volume or pressure overload, with hypertension usually causing pressure overload and either eccentric or concentric growth in the geometries of the heart [50]. During hypertension, cardiac hypertrophy usually progresses as a compensatory mechanism to maintain cardiac output despite increased wall stress. This adaptive cardiac remodeling advances into a decompensatory, maladaptive remodeling causing HF. This transition is not yet fully understood, although many cellular and molecular mechanisms have been proposed [51].

Although the transition from adaptive hypertrophy to pathologic hypertrophy is not fully understood, studies suggest that HSP22 may be involved in the development of HF from hypertrophy. One such study established HSP22 as a novel mediator of cardiac hypertrophy. First, the study showed that HSP22 is upregulated in a canine model of chronic cardiac hypertrophy resulting from pressure overload [36]. Second, they used primary cultures of 1-day-old Wistar rat cardiomyocytes as an in vitro cardiac cell model and used an adenovirus-overexpressed-HSP22 in cardiac myocytes. They found an increase in protein/DNA ratio and visual inspection of the cell size, indicating HSP22 overexpression stimulated cardiac hypertrophy. Third, they developed a transgenic mouse model to determine whether HSP22 was sufficient to allow cardiac hypertrophy to develop in vivo. Transgenic mice had increased heart weight/body weight ratios and larger cardiomyocytes, suggesting that HSP22 overexpression can cause hypotrophy in vivo as well. This study also gave insight into signaling pathways that HSP22 may be involved in. The results from transgenic mice demonstrated that HSP22 overexpression activates Akt and its downstream target p70S6K in a dose-dependent manner. It also showed a connection between HSP22 and the activation activity of the MAP kinase pathway and the subsequent upregulated glucose metabolism [36].

Although the above study demonstrates HSP22 sufficiency to drive the development of cardiac hypertrophy, the tendency of small molecular weight HSP to have some redundancy in function has been noted by others. Thus, it is important to determine HSP22’s function under cardiac overload. To assess this, an HSP22 KO mouse was subjected to transverse aortic constriction (TAC) to induce cardiac pressure overload. HSP22 KO mice develop a thinned and dilated left ventricle, leading to impaired contractile function and eventual mouse death. These results indicate that HSP22 plays an essential role in blunting the severity of cardiac pressure overload and slowing the eventual transition to heart failure [25]. Microarray analysis of both WT and KO mice exposed to TAC showed a differential expression of stress and inflammation-related genes, which are known to be controlled by the transcriptional factor STAT3, suggesting that HSP22 may influence this transcription factor to regulate gene expression. Subcellular fractionation and immunoprecipitation of cardiac tissues further demonstrated that the presence of HSP22 allows for the translocation and phosphorylation of STAT3 in both the cell nucleus and mitochondria. Oxygen consumption studies showed that deletion of HSP22 limits the ability of STAT3 to exert its physiological influence on cellular respiration. The same studies were conducted in vitro using cultured cardiomyocytes with HSP22 silenced and concurred with the mechanisms observed in HSP KO mice. It showed that HSP22 and STAT3 interaction was required for the activation of STAT3 via phosphorylation. Collectively, these results showed that HSP22 protects against heart failure in cardiac pressure overload by associating with STAT3 phosphorylation, leading to the expression of stress and inflammatory genes that appear to offer protection from heart failure [25]. On the other hand, hearts from HSP22 TG mice showed higher oxidative stress that increased cardiac hypertrophy and senescence [44]. Thus, in the context of oxidative stress, HSP22 overexpression will increase hypertrophy, leading to shorter life spans.

### 3.3. HSP22 in Ischemic Heart Diseases

Myocardial ischemic heart disease (MIHD), also known as coronary artery disease (CAD), caused by reduced blood and oxygen supply to the heart muscle, often leads to increased oxidative stress and myocardial hypoxia, or even death of the myocardium when the stimulus sustains [52]. Previous studies have shown that HSP22 was dramatically increased in cardiac cells in human and animal models in the condition of three major compensatory processes with MI, including myocardial stunning, myocardial hibernating, or ischemic preconditioning (IPC) [33,53]. These observations indicated that HSP22 is a crucial mediator of the protection of cardiomyocytes under ischemic stress and that it promotes cell survival under the hypoxia condition [17].

Studies in the cardiac-specific HSP22-overexpressing TG mouse demonstrated that overexpression of HSP22 improved cardiac cell survival in vitro and reduced infarct size in vivo [20]. Importantly, studies showed that HSP22 plays a protective role against ischemia/reperfusion with comparable protection to IPC and that it may also be mediated by similar signaling to IPC [20]. For example, it was found that HSP22 co-located with AKT and VCP, directly binding to these two proteins, activated transcriptional factor NF-kB, and STAT3 [34], which further induced (iNOS)—a well-known mediator of the second window of IPC. These data indicate that HSP22 could be used as a preemptory conditional mediator to induce a pro-survival mechanism, preventing cardiac injury due to ischemia/reperfusion.

In addition, with these exciting results of cardiac protection, the molecular mechanisms and the involved signaling transductions have been extensively studied to understand their underlying protective role. Several cell survival responses were observed in the ischemic hearts, including the regulation in ATP production by improving mitochondrial respiration, metabolism remodeling UPS function, and inhibiting ROS [17,18,20,54]. This relative information has been discussed in a recent review [33].

### 3.4. HSP22 in Age-Related Cardiomyopathy

In the current era, age has become a critical risk factor for prevalent cardiovascular disease. During aging and age-related cardiomyopathy, increased accumulation of lipids, fibrosis, and ROS production leads to cardiomyocyte stress and even death. In addition, progressive metabolic remodeling occurs during cardiac aging, which is inherently triggered and advanced by a series of alterations in autophagy and oxidative stress. All of these alterations result in the progressive decline in cardiac function during aging. Previous study showed that the heat shock response was diminished in the aged myocardium [55]. In addition, in the studies with Drosophila, it has been shown that overexpression of HSP22 extends the life span and increases resistance to oxidative stress [56], while it decreases lifespan in the absence of the mitochondrial HSP22 in Drosophila [57]. A recent time-course study demonstrated that, under physiological conditions, HSP22 played a vital role in maintaining cardiac function during aging. This study used aged knockout mice to assess whether loss of HSP22 hinders cardiac function and growth in aging animals. The deletion of HSP22 results in cardiac dilation starting from an early age and leads to progressive cardiac dysfunction with increasing age. Loss of HSP22 interrupted cardiac autophagy, metabolic remodeling, and oxidative stress, leading to ATP impairment and the activation of apoptosis in the heart, subsequently leading to declining cardiac function [26].

The mammalian heart is one of the most energy-consuming organs, and its function relies highly on timely and efficient energy metabolism. Although metabolic remodeling is a well-characterized hallmark of HF [58,59], the roles of energy metabolism in cardiac aging have not been fully understood. Results from a recent study showed that HSP22 deletion initiated the impairment of FA transport and FAO-related enzymes, undermining ATP production from FAO; this consequently stimulated glycolysis and gluconeogenesis to compensate for energy deficits. With increasing age, HSP22 deficiency was progressively exacerbated by the inhibition of glycolysis, particularly mitochondrial pyruvate utilization, which dramatically diminishes energy supply, leading to cardiac dysfunction and HF in HSP22 KO mice [26]. These results brought a new insight into the alteration in metabolism during aging—that mitochondrial FAO was impaired prior to the development of cardiac dysfunction, suggesting that promoting FA oxidation at the early stage might have a protective role against the progress of cardiometabolic disorder to the decline in cardiac function during aging.

### 3.5. HSP22 in Diabetic Heart Disease

Type II diabetes mellitus is a chronic metabolic disease characterized by the development of insensitivity to insulin [60]. Many diabetic patients acquire various cardiovascular diseases as comorbidities. Although the mechanisms underlying diabetic heart disease are not fully understood, the presence of endothelial cell dysfunction and injury in diabetes raises more attention in the pathogenesis of cardiomyopathy [24]. Chronic high blood glucose from diabetes precipitates endothelial cell damage, which can lead to cardiac damage [61]. Given the high prevalence of diabetics with cardiac damage and the perceived protective effects of HSP22, it is important to understand the consequences of HSP22 action in the diabetic heart.

A recent study demonstrated the protective role of HSP22 against diabetes-induced endothelial injury by inhibiting mitochondrial ROS formation [24]. The protective effects against hyperglycemia have been exhibited in both in vivo mouse models and in vitro. One study showed that when HSP22 is overexpressed in a mouse model of type II diabetes, the resulting endothelial injury from hyperglycemia is reduced. Results showed a reduction in hyperglycemia-induced endothelial injury, lower levels of the adhesion molecules ICAM-1 and VCAM-1 (part of the inflammatory response), and fewer pro-inflammatory cytokines present in the aortic endothelium of the mice. Similar results were found in HUVECs exposed to high glucose media. They further demonstrated that HSP22 does this by inhibiting mtROS. They exposed high glucose HUVECs to Mito-TEMPO, which is a mitochondrial-specific antioxidant to eliminate mROS. HUVECs exposed to MitoTEMPO, under a hypoglycemic background, had an increase in cell viability, less cytotoxicity, less abnormal mitochondrial morphology, good membrane potential, fewer cytokines, ICAM-1/VCAM-1 and monocyte adhesions, fewer mtROS (as determined per MitoSOX), and increased ATP production. However, when these studies were performed with HUVECs with silenced HSP22, the above improvements in cellular health and function were truncated. The results of this study suggest that HSP22 suppressed endothelial injury by eliminating hyperglycemia-mediated increases in mtROS. Furthermore, it showed that HSP22 maintained the balance of mitochondrial fusion and fission by mitigating mtROS in vitro. These data together indicated that HSP22 attenuated vascular lesions in a T2DM (type 2 diabetes mellitus) mouse model, which may be beneficial for the inhibition of pathogenesis of diabetic heart diseases by suppressing mtROS-mediated endothelial activation [24]. In addition, previous and recent studies have demonstrated that HSP22 participated in the glucose metabolism in the hearts under both physiological and pathological conditions [26,62], indicating a potential role in diabetic heart diseases. However, more evidence is needed by further investigation using a diabetic heart disease model.

## 4. Therapeutic Potential of HSP22 and Future Direction

Although the mechanism of H11K/HSP22 in different cardiac diseases is still poorly understood, the summarized information in this review still offers a promising therapeutic benefit for future studies.

First, in heart ischemia, pre-emptive conditioning by HSP22 leads to a new therapeutic strategy for patients under unstable angina and repetitive ischemic episodes or for those who will proceed with revascularization surgery. In addition to gene delivery [21], more investigations are expected in activating endogenous HSP22 under stressed conditions. Second, since excessive expression of HSP22 is associated with cardiac hypertrophy, utilizing drugs that inactivate the kinase activity of HSP22 may prevent cardiac hypertrophy, such as proteasome inhibitors [37]. Third, since molecular functions of HSP22 are highly dependent on BAG3 [39,40,41], therapeutic targeting of BAG3 has been considered as having potential in treating heart disease [63,64]. Fourth, studies have shown that recombinant HSPB8 or HSPB1 can interrupt amyloid formation, thus proving to be a potential therapeutic strategy for treating certain cardiomyopathies [32]. Inducers of small HSPs such as geranylgeranylacetone (GGA), which is a nontoxic anti-ulcer drug, have been shown to induce expression of HSPB8 and HSPB1, which in turn showed inhibition of interstitial fibrosis and reduction in heart size, therefore improving cardiac output and increasing survival rates of TG mice [33]. In addition, some HSP22-targeted drugs have also been reported to be beneficial in different diseases and aging. For example, atorvastatin was found to downregulate HSP22 expression in an atherosclerotic model in vitro and in vivo [65]. Teprenone was also found to increase HSP22 in response to oxidized LDL [66]. A histone deacetylase (HDAC) inhibitor trichostatin A (TSA) was also shown to extend the lifespan of Drosophila melanogaster through promoting HSP22 gene transcription [67,68]. Rhodiola rosea has been found to downregulate the mitochondrial HSP22 and improve human health and lifespan [69], thus presenting as a potential viable candidate to treat aging and age-related diseases in humans.

Although HSP22 shows promising therapeutic potential for cardiac diseases, many aspects regarding the role of HSP22 in the heart remain unknown, and the regulatory mechanism still needs deeper investigations. It is important to keep in mind that HSP22 exhibits a dose-dependent dual function. While the absence of HSP22 significantly reduces cell survival under stress, leading to a loss of protection against HF, long-term and high-level overexpression of HSP22 in mice increased oxidative stress and autophagy and reduced life span. In addition, overexpression of HSP22 provides the opposite results in ROS production between mouse model and Drosophila. In Drosophila, overexpression of mitochondrial HSP22 increases the lifespan [56]; paradoxically, HSP22-overexpressed TG mice show the opposite result [44]. The possible explanation is that in Drosophila, HSP22 is only in the mitochondria. In mice, cytosolic-produced ROS may contribute to the reduced lifespan [44,57]. Thus, the species difference needs more attention during the translational application. Investigating the regulatory mechanisms controlling endogenous HSP22 expression is also required, such as genomic or epigenomic regulation of HSP22 in the hearts may be needed through studies, as observed recently in HSP90 [48]. Furthermore, recent studies suggested that some extracellular heat shock proteins could be used as therapeutic targets and biomarkers in fibrosing interstitial lung diseases [70], which also brings a new direction to future studies of HSP22.

## 5. Conclusions

As summarized in Table 1, HSP is associated with various heart diseases through diverse cellular and molecular mechanisms. Dramatic upregulation of cardiac HSP22 expression has been observed in well-characterized protective conditions in animal models and humans. Overexpression of HSP22 was demonstrated to protect the heart against cardiac injury and to prevent heart failure in various forms of cardiac disorders, including acute and chronic myocardial ischemic injuries and pressure overload-induced cardiac hypertrophy [20,33,36]. Reciprocally, HSP22 deletion accelerates cardiac dysfunction and ventricular remodeling, leading to HF in the pressure overload-induced hypertrophic mouse heart [25] and in the aging heart. These studies together indicate that HSP22 plays a vital role in providing myocardial protection in the stressed heart.

## Figures and Tables

**Figure 1 cells-11-00114-f001:**
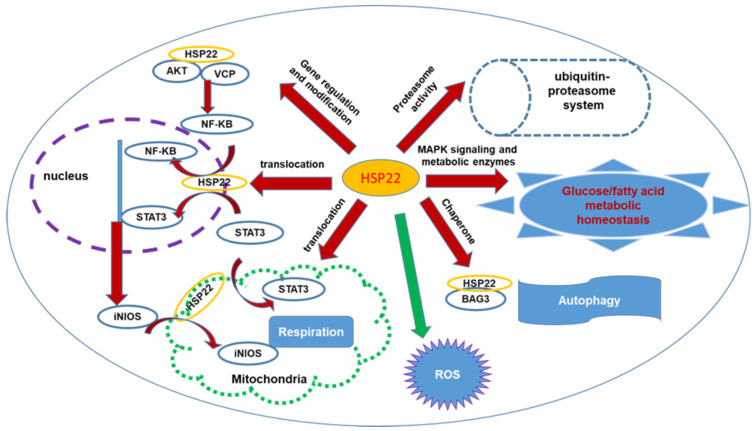
The illustration of the comprehensive signaling pathway and cellular processes regulated by HSP22 associated with cytoprotection in the heart. HSP22 interacts with AKT and VCP, inducing iNOS expression via activating NF-kB; it also facilitates multiple proteins’ subcellular translocation, such as STAT3, iNOS, modulating gene transcription in nucleus and mitochondrial function. HSP22 participates in activation of proteasomes regulating protein degradation and is involved in cardiac metabolism, ATP production, and ROS production and oxidative stress. HSP22 interacts with BAG3, modulating cardiac autophagy. Red arrows indicate increase or activation; green arrow indicates inhibition.

**Table 1 cells-11-00114-t001:** Summary of the association of HSP22 expression and function with heart diseases and cardiomyopathy.

Heart Diseases	HSP22 Expression and Function	Potential Therapeutic Targets
Dilated cardiomyopathy	Mutation of HSPB8(K141N) [48] HSP22 reduction with deficient BAG3 [26,42,43]	Increase HSP22 and BAG3
Cardiac Hypertrophy	HSP22 upregulated in cardiac hypertrophy [36]; HSP22 induces cardiac hypertrophy by activating proteasome activity [37]; HSP22 protects the heart against HF by activating STAT3/Akt signaling pathway [14,25]	Proteasome inhibitor STAT3 inhibitor
Ischemic Heart Disease	HSP22 upregulated in ischemic heart HSP22 protects the heart against ischemia injury by activating P13K/Akt/VCP/NF-kB/STAT3/iNOS [20,33,35]	Pre-emptive conditioning of Hsp22 Enhance iNOS mitochondrial translocation
Age-related Cardiomyopathy	HSP22 upregulated due to the intrinsic stress HSP22 deletion leads to cardiac dysfunction with increasing age by interrupting cardiac autophagy and impairing FAO) [26]	Increase BAG3, Enhance FA metabolism
Diabetic Cardiomyopathy	HSP22 upregulated HSP22 protect hearts through inhibiting mtROS) [24]	Reducing mtROS

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
