# Peer review of "Heat Shock Protein 22 in Physiological and Pathological Hearts: Small Molecule, Large Potentials"

_cells, 2021, doi:10.3390/cells11010114_

Round 1

Reviewer 1 Report

The minireview by Sun and Co-Authors would like to present the state of the art on the role of HSP22, also known as HSPB8 or H11 kinase, in cardiomyocytes and in cardiac diseases, such as hypertrophy, hischemia, heart failure.

The aim is clearly stated and outlined in the abstract, however each subsequent section is just a one-sentence summary of the present literature. None of the paragraphs enters deeply into the molecular or physiological details. The reader has to download and read each cited article in order to get this type of information. The complex behaviour of this kinase, which is also a chaperone, is never really outlined. The contradictory results from the literature are never in depth discussed.

The two figures are very schematic diagrams listing the same keywords of the text without giving a real extra information. The arrows are not causative links nor logical consequences.

The Authors have cumulated a list of sentences without really explaining what they anticipated into the abstract. The impression is that they have put in words the slides of one seminar in which a list of briefs sentences had been given (and some errors in the punctuation are a symptom of this).

Reviewer 2 Report

Sun et al. extensively review and summarize the current literature on Heat shock protein 22 in physiological and pathological hearts. Given the protective role of HSP22 against various cardiomyopathy, such as diabetes, ischemia, heart pressure overload, and aging. This aspect of HSP22 regulation is key to develop therapeutic options in various heart diseases making this review highly important to the field. 

While the topic is very relevant, the manuscript requires better organization. Such as using the sub-headline. For example, in “2. The molecular mechanisms of HSP22’s cytoprotection in cardiomyocytes.“, adding the highlight of the molecular mechanisms 2.1, 2.2, … in each topic can help readers to better understand the flow.

I have a few suggestions for the aspects of HSP22 regulators that should be discussed. 

  1. What is the upstream of HSP22. What regulate HSP22 expression in heart? Are there any post-translational modification involved in HSP22 expression level? Are regulators of HSP22 related to heart diseases?
  2. Small heat shock protein HSP22 interacts with a lot of chaperone protein. If they are important regulators, they need to be discussed in the article.
  3. In figure 1, what directs HSP22 to go to different pathway (the blue arrows)?
  4. I found some difficulty to comprehend figure 2. Could you provide a summary table with different category of HSP22 mutation(nucleotide)/overexpression/downregulation; affected pathway; affected proteins (enhance or reduce activity); associated heart disease; found in animal model or patients; reference 
  5. Could you also make therapeutic targets into a table?

Formatting issues

  1. A typo HSP2 in 318
  2. Spelling issue of “chaperone” in 11 and 79
